# On the Road Together: Issues Observed in the Process of a Research Duo Working Together in a Long-Term and Intense Collaboration in an Inclusive Research Project

Sofie Sergeant [1,2,*], Henriëtte Sandvoort [3], Geert Van Hove [4], Petri Embregts [5], Kim van den Bogaard [6], Elsbeth Taminiau [5] and Alice Schippers [1,7]

1 Disability Studies in Nederland, 3453 NW De Meern, The Netherlands; alice.schippers@disabilitystudies.nl
2 Pabo, SvO & Youth Studies, University of Applied Sciences, 3584 CH Utrecht, The Netherlands
3 LFB, 3527 GV Utrecht, The Netherlands; info@lfb.nu
4 Department of Special Needs Education, Ghent University, 9000 Gent, Belgium; geert.vanhove@ugent.be
5 Tranzo, Tilburg School of Social and Behavorial Sciences, Tilburg University, 5037 DB Tilburg, The Netherlands; p.j.c.m.embregts@tilburguniversity.edu (P.E.); elsbeth.taminiau@gmail.com (E.T.)
6 Koninklijke Kentalis, 5271 GD Sint-Michielsgestel, The Netherlands; ki.vandenbogaard@kentalis.nl
7 Disability Studies, University of Humanistic Studies, 3512 HD Utrecht, The Netherlands
* Correspondence: sofie.sergeant@hu.nl

**Abstract:** Inclusive research practices can lead to progress towards an inclusive society. With this study, we aimed to gain insight into dilemmas and catalysing processes within the long-term collaboration of an inclusive research duo: one non-academic researcher who lives with the label of intellectual disabilities and visual impairment, and one academic researcher. Both researchers kept personal diaries about their collaboration process. Inductive thematic analysis, individually and as a group of authors, was employed. Our findings reveal six necessary conditions for diversity-sensitive work in inclusive research: (a) experiencing belonging within the research group, (b) empowering people in a team through growing self-awareness and competence-building, (c) having room for reflection and searching for various ways of communication, (d) sharing power and ownership of research processes, (e) having enough time to foster the above conditions, and (f) joining in a mutual engagement in accommodating vulnerability in dialogue and collaborative work. Awareness of stigma-related issues and the risk of tokenism is also required.

**Keywords:** collaboration; inclusive research; intellectual disabilities

## 1. Introduction

The way society views people with intellectual and developmental disabilities (IDD) is shifting—values such as inclusion and empowerment have become more important (Zaagsma et al. 2020). In research, co-creation and collaboration with people with IDD are sought in response to this change, and to provide the evidence and tools required to foster societal inclusion. Working together with the people the research concerns is framed as "inclusive research" (Sergeant 2021). Advocacy of inclusive research practices is found prominently in the field of intellectual disability research (Nind 2014), but co-researchers are often unsure of how to proceed, and frequently encounter problems with communication, ownership and control, and equitable partnership (Bigby and Frawley 2010). Inclusive research contributes to the quality of both the process and outcomes of research when it helps to recognise, foster, and communicate the contributions of people with intellectual disabilities, and when it endows better lives for the wider population of people with intellectual disabilities (Walmsley et al. 2018), "to redress wrongs, both past and present" (Nind 2014, p. 15). However, research conditions should be considered to

ensure quality, prevent tokenism, and protect the well-being of all people included in the research team (Bigby et al. 2014; Chapman 2014; Strnadová et al. 2014).

The authors use the definition of inclusive research of Walmsley and Johnson—"research in which people with learning disabilities are active participants, not only as subjects but also as initiators, doers, writers and disseminators of research" (Walmsley and Johnson 2003, p. 9)—as a basis for their research projects. Walmsley and Johnson (2003) state that inclusive research (1) must address issues that really matter to people with intellectual and developmental disabilities, (2) must engage in research that ultimately leads to improved quality of life for them and their families, (3) must access and represent their perspectives and ambitions, and (4) must take place in a research community that treats people with IDD with respect. This fourth principle implies that inclusive research must be built on respectful collaboration between people who have scientific knowledge and people who have more practical, experience-based knowledge. This is not always simple: we agree with Chalahanová and colleagues that "time is needed to relax into relationships that are allowed to build slowly and organically" (Chalahanová et al. 2020, p. 155).

Prompted by the United Nations Convention on the Rights of Persons with Disabilities, inclusive research has gained increased attention (Embregts et al. 2018). However, good intentions are not enough to conduct inclusive research in practice. With this article, we have engaged with Nind's call for critical self-reflection and shared reflection within the field of inclusive research (Nind 2014). We aim to provide a clear account of what happens within our long-term collaboration, and why and how we engage in collaborative production of knowledge between an academic and a person who was traditionally thought of as a participant or object of study (Duggan 2020; Frankena et al. 2018). Acknowledging that developing an equal relationship throughout the research process is a crucial departure point for true collaboration (Duggan 2020; Embregts et al. 2018; Frankena et al. 2018; Nind 2014), this article presents in-depth research into the process of building an equal relationship.

The central research question of this study is:

What themes, problems, and processes are observed in the process of a research duo (one non-academic researcher who lives with the label of intellectual disabilities and visual impairment and one academic researcher) working together in a long-term and intense collaboration on an inclusive research project in which they developed, organised, and delivered training to inclusive research projects in The Netherlands? This has been concretised in the following sub-questions:

- How have the researchers experienced their intense collaboration?
- What were the advantages and added value of inclusive practice?
- Which struggles and oppositions did they encounter within their own collaboration, and in the wider context?
- How did they deal with these challenges, and what is the value of their solutions for future inclusive research?

The research duo developed a training and coaching package for inclusive research teams within the context of a larger four-year inclusive research project, "Working Together, Learning Together" (WTLT). The research duo collected questions and needs for training from ten inclusive research projects in The Netherlands. Based on literature research and the information collected, researchers sought to learn how to develop and provide training and coaching to inclusive teams. The training development process was designed in an action-oriented and iterative spiral of learning, building and creating modules, evaluating and reflecting, adjusting the training modules, etc. The development of this training has been described in Sergeant et al. 2020. We learned from international research (Nind 2014; Nind and Vinha 2014; Strnadová et al. 2014) about the importance of training, team building, talking things over, and collaborative reflection. The training was developed because training and coaching for inclusive research teams were not available in The Netherlands (Sergeant 2021), despite increased calls for inclusive research, including from Dutch and international funding agencies.

The data on which this article is based were collected while the inclusive research duo created the training together (Sergeant et al. 2020). In this iterative process, the research duo shared all tasks, explored what this collaboration needed, and reflected on the themes and problems they encountered on the road to jointly developing and organising the training. Providing insight into the critical reflection process of a research duo enables a deeper understanding of themes, dilemmas, problems, and catalysing processes involved in working closely together. By documenting this work, we aim to inspire and support future inclusive research projects.

## 2. Methods

From 2016 to 2020, the seven authors conducted a nationwide inclusive research project in The Netherlands. For this project, we were asked by The Netherlands Organisation for Health Research and Development (ZonMW) to bring together questions and needs regarding inclusive research from 10 Dutch research projects (Sergeant 2021). Based on the questions and needs gathered, we started creating a training programme for inclusive researchers through an iterative and inclusive research process. Therefore, WTLT employed action-oriented qualitative research methods. In action-oriented research, "emphasis is placed on producing knowledge that can be used by community partners to contribute to positive social change and the well-being of individuals, families and communities" (Small and Uttal 2005, p. 938.) Our research work involved a reflective practice of developing training for other inclusive research teams, continually building on findings. We used action-oriented research, because we aimed to catalyse—on the road—positive change through creating time and space for training (Kidd et al. 2017), and by supporting trainers and participants to become more reflective in their work and collaboration.

The study spanned four years, as the research duo worked closely together and collected data in various forms. The Medical Ethics Review Committee of VU University Medical Centre (FWA00017598) confirmed that the Medical Research Involving Human Subjects Act (WMO) did not apply to this study and approved this study. This article focuses on this long-term collaboration between the first author as the academic researcher (Researcher 1) and the second author as the researcher who lives with the label of intellectual disabilities and visual impairment (Researcher 2). We refer to this pair of researchers as "the research duo" in this article. Researchers 1 and 2 did not know each other prior to the study. The researchers were paired on this project because Researcher 1 has experience working alongside people with disabilities and because Researcher 2 has experience in contributing to inclusive research projects. In Figure 1, the inclusive research duo is presented. The photo is a still – printed with consent of both researchers—from the film the duo made to introduce their collaboration and research project on developing training for inclusive research teams. The film can be downloaded from https://youtu.be/pOT2iRiEps4 (accessed on 14 January 2022).

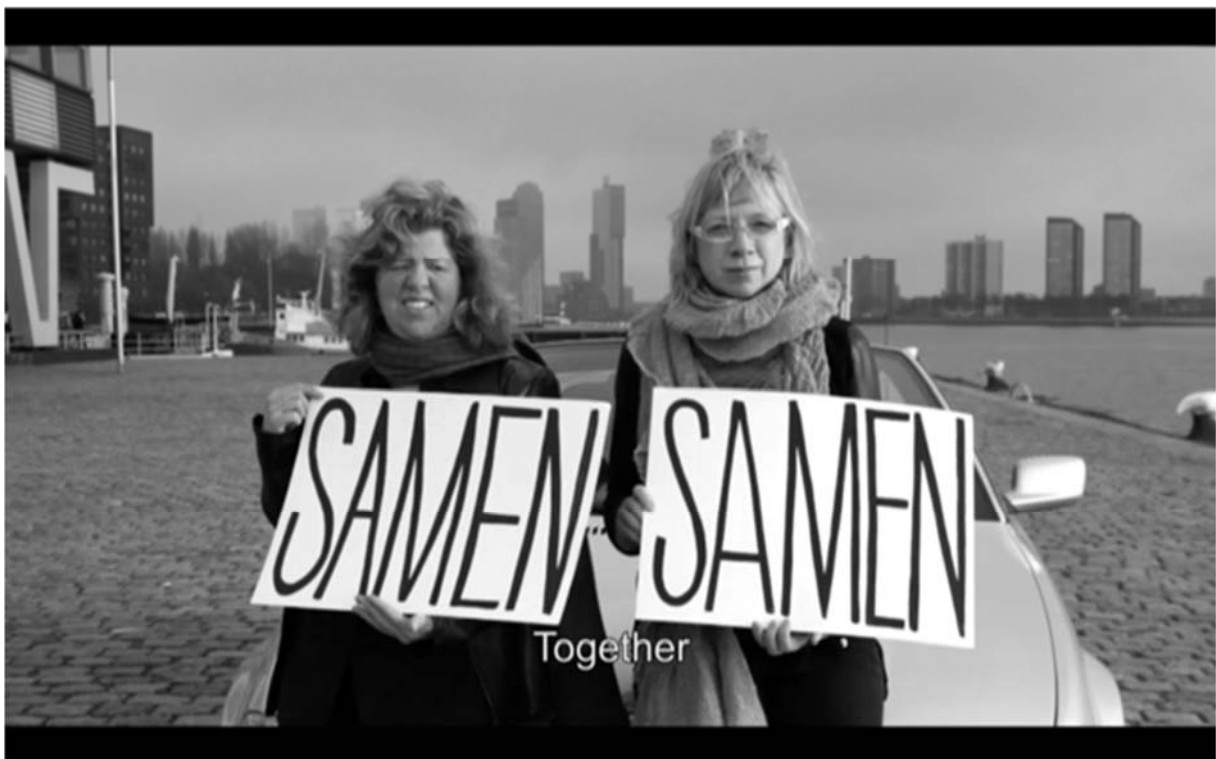

**Figure 1.** Inclusive research duo.

### 3. Study Design

The research duo worked together to develop and give a package of training and coaching sessions. During this process, the research duo collected data through participant observation. In participant observation, researchers are involved in the setting under study as both observer and participant (Maso and Smaling 1998; Reason and Bradbury 2001). Participant observation helps to identify and guide relationships, to learn about interaction, to examine how things are organised and prioritised in a setting, and to learn what is important to people (Kawulich 2005).

Based on these participant observations, the research duo wrote reflections in the form of extensive fieldnotes in individual research diaries at least two times per week, for four years. The research question and sub-questions served as starting points for our writings in the research diaries. As Bolger and colleagues state, "In diary studies, people provide frequent reports on the events and experiences of their daily lives. These reports capture the particulars of experience in a way that is not possible using traditional designs" (Bolger et al. 2003, p. 579). For this diary study, the research duo embedded an event-based design. A diary report was entered directly after every interview, training, or meeting connected with the research project. This event-based design was appropriate because the process of working together included triggering events—events that the research duo experienced as very positive or very negative—and attention to these could unfold dynamic phenomena (Bolger et al. 2003, pp. 590–91).

The research duo also decided to share several stories from these diaries in 35 published blogs and vlogs, which also form part of our research data. The blogs and vlogs were created based upon the diary notes. One additional blog was written by a journalist after he interviewed the research duo on why the research duo worked together, on what the duo encountered in their joint work, and on how they dealt with hindrances and challenges (Lingbeek 2017).

This process of creating diary notes provided a way to document our research journey and an early opportunity to critically reflect on experiences not long after they had occurred. The rationale for deciding to create blogs and vlogs based on the diary notes was to

support our action research goals, to make our research work and results more visible and accessible to a broader public, and to be more transparent about the methods and process of our study (Hookway 2008; Mortensen and Walker 2002; Reason and Bradbury 2001). The blogs and vlogs included texts, images, and films to reach populations otherwise geographically or socially removed from the researchers (Hookway 2008). All blogs and vlogs were published (in Dutch) on the site of Kennisplein Gehandicaptensector (https://www.kennispleingehandicaptensector.nl (accessed on 14 January 2022)), the Dutch online knowledge exchange platform on disability, inclusion, quality of life, care, and support. The diary notes, blogs, and vlogs were created during the four phases of the research project: (A) literature research and introductory meetings, (B) exploration of needs and gradually building up the training program, (C) expanding the team and involving more research projects, and (D) adjusting and reorganising the training (Sergeant et al. 2020). Every quote used for this manuscript is assigned to Researcher 1 or Researcher 2 and to a research phase A, B, C, or D in the Findings section.

In summary, the participant observations resulted in diary notes created—separately—by both researchers. Some of these diary notes were processed by the research duo and published as blogs and vlogs. The process of formatting and publishing the blogs and vlogs was facilitated and supported by Kennisplein Gehandicaptensector. For this article, all the quotes from the diaries and blogs are translated by first author, supported by a native English speaker.

The present study was conducted through teamwork, which was valuable in helping us to assess the study from a more critical perspective and to recognise and address its limitations. To deal with the threats to validity (Robson 2002)—referring to the integrity and application of the methods undertaken and the precision with which the findings reflect the data (Noble and Smith 2015)—we applied the following strategies. Firstly, as already stipulated, the large team of researchers (all authors) provided valuable feedback and suggestions for improvement. Secondly, the research work was conducted over four years: this prolonged involvement yielded a large amount of data, retrieved in various settings and situations, over four research phases (Sergeant et al. 2020). The third strategy is triangulation: we used different instruments of data collection, as explained above.

## 4. Analysis

Qualitative data consisted of diary entries (field notes from participant observants), and the online blogs and vlogs, which included film and photos as well as text. A thematic analysis was conducted. Thematic analysis is the process of identifying patterns or themes within qualitative data (Braun and Clarke 2006). Four steps were taken during this process.

First, all research data materials were printed, listed, and numbered. Every researcher in the team of seven authors received a package of data. Materials were divided and shared in a way that ensured analysis of every piece of raw data by a minimum of two people, to integrate different perspectives and interpretations.

Second, the researchers familiarised themselves with the data by repeated reading or viewing, searching actively for meanings and patterns (Braun and Clarke 2006). Once the researchers had familiarised themselves with the data, they engaged in coding the data, identifying important sections of text, and attaching labels to index them (Braun and Clarke 2006). In this second step, the data were open-coded, with data fragmented and titles assigned using short terms and phrases. Through this inductive thematic content analysis phase, individual researchers were asked to document theoretical ideas and reflections developed through immersion in the data, including values, interests, and growing insights (Lincoln and Guba 1985). The aim of this step was to stay as close to the content as possible and to guarantee authenticity. Researcher 2, who already had extensive experience in coding in other research projects in the past, used accommodations during the analysis phase based on her needs, including a computer screen magnifier and large print copies of text.

Third, the research group gathered in the same room with their code lists and notes. The aim of the meeting was to code axially, whereby the open codes that belonged together were sorted under a theme (Braun and Clarke 2006). The group used Post-it notes on a wall to shape an overview of the identified themes and to support the search process. In the joint meeting, the findings of Researcher 2 were shared first and guided the whole group, providing structure throughout the meeting. It is important to note that this research group has worked together on a long-term basis. Therefore, the atmosphere was collegial, while still being intense and critical. This process of joint analysis was filmed.

In the last step, coding was selective, determining the relevance and coherence between themes (Braun and Clarke 2006). The first author used the film of the joint analysis meeting, the photos of the Post-it wall, all the different code lists, and notes of the researchers. The first author sent the results of this selective coding process to all authors and received feedback. We tried to ensure that the academic's point of view did not silence the perspective of the non-academic researcher by using the original coding documents of all researchers involved, and by re-watching the group analysis process together with Researcher 2. The comments given by Researcher 2—without the others being present—were also included in the analysis process. This process took months to come to a structure that was satisfying to all involved in the analysis process. The result of this inductive analysis process is the backbone of this article.

## 5. Findings

In this section, we present the results of the analytical process. Through the collaborative analysis process, we inductively derived a manageable structure of six significant themes:

1. Belonging
2. Self-awareness and competence-building
3. Communication
4. Sharing power
5. Time
6. Vulnerability

Although we reached consensus that data cohered together meaningfully within these themes, and we agreed that there was a clear distinction between themes (Braun and Clarke 2006), we also noticed links between themes. These links are elaborated in the Discussion section.

In the Findings section, we embed extracts of raw data from field notes/diaries, blogs, and vlogs to illustrate the complex story captured in the data, to do more justice to the richness of the data rather than to provide only a flat description (Braun and Clarke 2006).

### 5.1. Theme 1: Belonging

In their personal research diaries, the research duo frequently wrote on the theme of belonging and how this is impeded because of prejudices and (self-)stigma. As Researcher 2 told a journalist:

> *"At a certain point, you know that society is like that. I know no better than people looking at me or staring at me. That happens, I can't see that because of my visual disability. But my researcher colleague sees that and gets angry about it".* (Lingbeek 2017)

The theme of prejudice and (self-)stigma was often elaborated upon as an important barrier for belonging in research and in society, something that stands in the way of equal cooperation. On this theme, Researcher 1 (R1) and Researcher 2 (R2) wrote in their diaries:

> *"People speak to my colleague with a high [childish] voice. Sometimes people speak to me and ignore her. My colleague says she is used to this . . . She wonders about me getting upset by this".* (R1—phase B)

> *"This research is very confusing to me. My whole life people say to me that I do not know. And now my research colleagues tell me that I do know. That I should take more*

*initiative. That they don't have the answers too. When I think about this, I experience it as a compliment. But it is confusing anyway".* (R2—phase A)

While meeting other inclusive research teams, the research duo witnessed hierarchy-based dynamics. They saw people struggle to not be seen as the most disabled one in the room. The research duo observed people debating about who was the best expert by experience, which is illustrated by this quote from the diary of Researcher 1:

*"We witnessed discussion and quarrels today between experts by experience on who is the best researcher? They concur about who is the most attributed and able to contribute to the research. This battle reveals—I think—their hard work to belong to the research project, doing their ultimate best to succeed".* (R1—phase C)

The research duo talked about this experience. They learned how important it can be to organise reflection and dialogue on (changing) responsibilities in their own collaboration, leaving the research project, quitting, and taking up less or more work. They decided this should not occur in a way where one person feels disrespected or that their work is not valued, but both should feel that they can share ambitions or place limits and can decide for themselves whether to (temporarily) quit or continue.

We connected these results with the concept of belonging, referring to the definition of inclusion that means not only "taking part", but also having rights and responsibilities as a legitimate member of a group (Van de Putte et al. 2018). A group can exist at different levels, from the macro level (society as a whole) to at the micro level (in this case a research group): people may (not) feel part of "regular" society and/or might (not) experience belonging within the research group.

*5.2. Theme 2: Self-Awareness and Competence-Building*

Self-awareness and competence-building were catalysed for the research duo. Researcher 1 wrote in her diary about her encounter with her colleague, Researcher 2. In this encounter, Researcher 2 explained that while working in the research context, she realised that she is more than "a disabled person" and that she did not want to narrow her work down to just the disability experience:

*"My colleague told me today she doesn't like the idea of working for a self-advocacy movement run by people with ID anymore. She started feeling uncomfortable because—in her job—each time she must introduce herself as a person with ID".* (R1—phase D)

Acknowledging and valuing differences in perspectives, experience, and knowledge come forward as vital elements in inclusive research. At the same time, lack of education was reported by Researcher 2 as a burden and cause for frustration. The following quote from Researcher 2's diary discusses being excluded from regular education because of her disabilities, while being involved in inclusive research has allowed her to develop talents and skills:

*"I am not happy with the education I had. I wished I had gone to an inclusive school. I never had the opportunity to do the studies I aspired to. Now I am happy with the opportunity to learn on the job and to contribute to research".* (R2—phase D)

The intense and long-term collaboration inspired Researcher 1 to think about her own disability experience:

*"Growing up with a grandmother with major psychological problems and having a daughter with a metabolic disease, I begin to realise that these life experiences have contributed to a deeply rooted awareness of inherent complexity and entanglement of life experiences and knowledge".* (R1—phase D)

*5.3. Theme 3: Communication*

For both partners to join the research work, sometimes other methods of communication were needed considering the visual impairment of Researcher 2, as depicted in a diary quote by Researcher 2:

*"From the introductory meetings we have included our notes in an (online) Prezi presentation (prezi.com) . . . Because I have a visual impairment, this helps me a lot . . . We have an overview of the meeting . . . And at the same time, I can make one section of the presentation much bigger".* (R2—phase A)

Collaboration with visual artists, photographers, and filmmakers was found to be an indispensable condition for the research duo, as illustrated in a blog written by Researcher 2 about a film the research duo created together with filmmakers in phase A to introduce their research work to a diverse public (see also Picture 1, a still from this film):

*"In this film two tough women are driving a Cabrio. The film has the appearance that they get the job done together and are on the road together for this. Under all circumstances! And we do that too... My colleague and me. Before we made the film, we looked for what binds us: we love good music, travel, the feeling of freedom . . . and in the Cabrio that all comes together".*

We could show this film to students, to professionals, to researchers, to experts by experience, and their families. They all could grasp the essence of our message: we try to collaborate, and that is interesting, sometimes difficult, and always far away from the "pity discourse".

Learning from these experiences, the research duo created a film (https://www.youtube.com/watch?v=wStYLc1a7-Y (accessed on 14 January 2022)) revealing what they have learned from their research work instead of choosing for an "easy read" article. We believe that this makes the research results more widely accessible: no reading skills or large investments in time, energy, and focus are needed.

Thus, from our data, we learn that sometimes, other communication modes are needed because of the researchers' impairments, or because of participants' needs, but also to make the research results more widely accessible to a broader public.

*5.4. Theme 4: Sharing Power*

The theme of power in the research work focuses on how decisions are made, who is in control, and who has influence in the research process. The research duo learned that in every phase of the research work, they had to keep searching for their (changing) roles and responsibilities. The quote below goes back to the start of the WTLT research project. Researcher 2 asked Researcher 1 for "the next step". Researcher 1 wrote in her diary:

*"It was as if Researcher 2 asked me to give her the answers. And I do* not *have them. It is as if she asked me to give homework. This is not how I want to work together".* (R1—phase A)

This incident was crucial for shaping the research duo's collaboration. Long talks and many hours of collaborative reflection were needed to work this out for both researchers. Both researchers felt like they kept on profiting from this incident. The academic researcher said, *"I don't know"*, which brought confusion, but also space for Researcher 2 to take more power and control.

Every research project starts with decisions about the focus of the research question and the design of the research. In our research project, Researcher 2 had a decisive role in this phase. She contributed to setting the research agenda, designing the research process, and deciding where the money goes. In her diary, Researcher 1 wrote the following on fostering shared power and ownership in inclusive research:

*"If we want researchers to design and write projects in co-creation with experts by experience, grant-giving organisations will need to provide the necessary time to co-create and co-write. The grant-giving organisations will also have to acknowledge that predictions on the used methods and the timeline are more difficult to make if you collaborate with experts by experience. Some extra space for adjusting time and method to the needs of the team will be appropriate".* (R1—phase C)

Ownership of the research and the research question was of major importance in the research duo's collaboration. Both researchers were eager to realise the goals of the research. This helped to motivate them during their four years of intense collaboration.

*5.5. Theme 5: Time*

This theme is strongly intertwined with all the other themes but proved to be an important condition in itself. Researcher 1 wrote, after she met Researcher 2 for the first time:

> *"Our first date took place in my house. After a long day talking and getting to know each other, my new colleague says to me: 'I know what you need. You need structure. And I am able to give you this.' I smiled. Ouch. She already recognises something that is very true. Structure is what I need; and I need somebody else to help me create it".* (R1—phase A)

Having enough time to get to know each other surfaced as a major issue during our research. The duration of the research project can catalyse ownership of the project and ambitions to evoke positive change through research. The research duo shared thoughts on the cruciality of creating enough time: to get to know each other, to discover what the other needs, and to take up roles in the project that fit the temperament, competences, and ambitions of the researchers.

*5.6. Theme 6: Vulnerability*

When the research duo started delivering training to inclusive teams, they reported feeling insecure. They had many questions about how to enter these teams and how to position themselves. As Researcher 1 wrote:

> *"We felt being watched. We had to be good; we felt like we were not allowed to make mistakes".* (R1—phase A)

However, this changed over time:

> *"Now we feel more relaxed in the cooperation: the cramp disappeared. Vulnerability is an important issue: can we be vulnerable; can we make mistakes and learn from that?"* she later wrote. (R1—phase C)

The research duo struggled in the beginning of their collaboration with mutual engagement in dialogue. Researcher 1 admitted in her field notes that she was used to taking care of people with disabilities, instead of working with them as colleagues, giving feedback and sharing thoughts. This is illustrated in the next quote from her diary:

> *"How must I share my thoughts with my colleague? How can I bring in my questions, insecurities, and delicate thoughts on our collaboration? I am afraid to hurt her feelings".* (R1—phase A)

The research duo learned that admitting to themselves and others that they were constantly struggling and searching was very helpful in their collaboration. This process brought relief and tranquillity to their relationship and their research work.

## 6. Discussion

In our research, it became clear that if people with disabilities and their colleagues become aware of their knowledge, their power, and the danger posed to their collaboration by (self-)stigma (Scior et al. 2015), something changes in their lives and in their collaborations. The research duo started their work together with a binary vision, juxtaposing the academic researcher and the non-academic researcher living with disabilities. On the road, they discovered how entangled their lives are, how Researcher 2 became a researcher with academic skills herself, and how Researcher 1 came to reflect more on her own life story and experience.

For all research members to be able to flourish and to develop talents, a diversity-sensitive context (MacDonnell and Macdonald 2011; Peels and Sergeant 2018) must be

created, with training and coaching provided as needed. On the one hand, this context must provide support and protection for all members of inclusive research teams, and on the other hand, it must presume and support competences of all team members: both have been identified as very important pre-conditions for inclusive research (Embregts et al. 2018; Strnadová et al. 2014).

An overarching theme in our findings is related to stigma. Scior et al. (2015, p. 15) define stigma as "the co-occurrence of these stigma components: labelling, stereotyping (that is negative evaluation of a label), prejudice (that is endorsement of negative stereotypes), which lead to status loss and discrimination for the stigmatised individual or group." During an intense collaboration in which the researchers almost did everything together, the research duo became aware of the impact of prejudice, (self-)stigma, and (their own) binary thinking, which influence not only the position of people with disabilities in our society, but also their role in research. The label of intellectual disability can cause low self-esteem (self-stigma), lower expectations in society, and being positioned lower on the research participation ladder (Arnstein 1969; Kliewer et al. 2015; Tritter and McCallum 2006).

Our research data reveal that more participation is not always better. What counts is meaningful participation, in which all participants are convinced of the importance of the contribution of all involved, and of the worth of each other's knowledge to the team. Rather than being asked to participate for instrumental or tokenistic reasons, people with IDD should be seen as bringing more diversity, quality, and richness to the research process and (dissemination of) products. This means time for reflection on inclusive collaboration is crucial (Bigby et al. 2014; Johnson and Johnson 2009; Strnadová et al. 2014): open and sincere dialogue and reflection help greatly and can form a solid basis for daily work. Our data show the importance of collaborative reflection on the meaning of disability, on what people need within a research collaboration, and on the experience and impact of prejudice and (self-)stigma during inclusive work processes. It is only possible to discuss this if there is time for slow research, if there is shared history and reciprocal trust. Attention must be also paid to matching talents, roles, and tasks in the research process (Embregts et al. 2018; Frankena et al. 2018; Nind 2014).

## 7. Future Research

The authors recommend further investigation into how inclusive research in a disability-focused context and the conditions associated with its success can be a catalyst beyond research that is inclusive of disabled people, leading towards a less hierarchical academic world and a more supportive, democratic, and safe space for researchers. This is especially important in IDD research, because the long history of excluding and disempowering people with an IDD label from research processes has likely impeded progress and may at times have contributed to abusive practices.

People with intellectual disabilities and people with mental health problems are often seen as the lowest in the disability hierarchy (Deal 2003; Scior and Werner 2016). This hierarchy perpetuates the notion that some disabilities are more acceptable than others in our culture. This hierarchy can be internalised and deployed by people with disabilities, as well as by those without. From the data, we observe that the disability hierarchy appears to be a barrier to inclusive research. Future research is needed to gain deeper insight into this process.

Currently, many organisations provide funding on the condition that leading researchers work closely together with experts by experience. However, tokenism can still lurk within such constructions, along with the risk of "data robbery": stealing the stories shared by experts by experience without acknowledging their ownership (Embregts et al. 2018; Nierse and Abma 2011; Nind 2014). The authors recommend further research to gain insight into what conditions must be met to ensure that research participation and co-creation deliver on their promise. Based on our findings, we would like to encourage future researchers to engage in critical reflection on their own collaboration process, frankly writing on both the richness and the struggle that comes with inclusive research.

Changes made to the Dutch research funding environment following The Netherlands' ratification of the UN CRPD have fostered more inclusive research, but in The Netherlands, as elsewhere, funding mechanisms continue to disadvantage disabled researchers, and especially those with IDD who may need additional funded support to fully participate. For example, non-disabled researchers are more likely to earn a salary for their research, while disabled co-researchers may be relegated by funding or disability benefits rules to no or very low renumeration. This impacts the potential of inclusive research, including ours, to be truly "equal". Research environments continue to valorise production over contribution and effort, which can also disadvantage researchers with IDD who expend considerably more effort to contribute. In our research process, we worked hard to manage financial equality as well as intellectual property. Both researchers were paid and were involved in every phase of the research process, being valued in their expertise. We realise some inclusive research teams work in different conditions, and are struggling with payment and time pressure. So, based on our results, our final advice for further research is to rearrange the way of funding.

## 8. Limitations

We recognise the limitations of analysis based upon the experience of one research duo in The Netherlands. As mentioned earlier in the Methods section, prior to this study, Researcher 1 already had experience working alongside people with disabilities and Researcher 2 already had experience in contributing to inclusive research projects. Nevertheless, because the research took place during a process that involved close contact with multiple inclusive teams, we believe we have identified some critical contextual factors which are crucial for inclusive research and for collaboration within teams.

An important factor in the research duo's communication was that, with support and specialist equipment, Researcher 2 was able to keep a reflective diary and contribute to blogs and vlogs. Some research participants with IDD would need more support to reflect on their experiences and act as self-advocates within a research setting; research participants who are non-literate or non-verbal would need different forms of support to contribute and to document their contribution.

## 9. Conclusions

In this research, we documented and explored the issues, observed in the process of a research duo working together in a long-term and intense collaboration on an inclusive research project (WTLT) in which they have developed, organised, and delivered training to inclusive research projects in The Netherlands.

Our findings reveal six necessary conditions for diversity-sensitive work in inclusive research: (a) experiencing belonging within the research group, (b) empowering people in a team through growing self-awareness and competence-building, (c) having room for reflection and searching for various ways of communication, (d) sharing power and ownership of research process, (e) having enough time to foster the above conditions, and (f) joining in a mutual engagement in accommodating vulnerability in dialogue and collaborative work.

Within our research results, one could identify a "true contradiction" (Rieger and Young 2019): (a) Inclusive research needs a well-prepared diversity-sensitive research environment, but at the same time, (b) "fear of doing it badly should not prevent us from attempting it" (Sin and Fong 2010, p. 21). Struggle is central in inclusive research, and therefore, we believe we are all responsible for welcoming this experience of negotiation and transformation, and discovering what Melanie Nind (2014, p. 84) means by "the full potential of inclusive research".

**Author Contributions:** The first and second author contributed equally to this article. In the preparatory, analysis and writing phase of the research, all authors were involved. All authors have read and agreed to the published version of the manuscript.

**Funding:** This research project was funded by ZonMw within its national program for people with disabilities, 'Gewoon Bijzonder' ('Typically Special').

**Institutional Review Board Statement:** The Medical Ethics Review Committee of VU University Medical Centre (FWA00017598) confirmed that the Medical Research Involving Human Subjects Act (WMO) did not apply to this study and approved this study.

**Informed Consent Statement:** Informed consent was obtained from all subjects involved in the study.

**Data Availability Statement:** The data that support the findings of this study are available on request from the corresponding author.

**Acknowledgments:** We thank ZonMW for their trust and support. The authors also want to thank Mitzi Waltz for reading the text closely and giving it a final check as a native English speaker.

**Conflicts of Interest:** The authors declare no conflict of interest.

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
