# Peer review of "On the Road Together: Issues Observed in the Process of a Research Duo Working Together in a Long-Term and Intense Collaboration in an Inclusive Research Project"

_socsci, doi:10.3390/socsci11050185_

Round 1

Reviewer 1 Report

Thank you for giving me the opportunity to review socsci-1577247, “On the road together: Issues observed in the process of a research duo working together in a long-term and intense collaboration in an inclusive research project.” A better understanding of what it takes to work together on research with a co-researcher with an intellectual disability is very valuable and I commend the authors for making their experiences explicit and set preconditions to work together in future research. Still I have some minor suggestions/queries.

Introduction

  • Overall, the introduction is clearly written. At the end of the first paragraph, the researchers stated that ‘Inclusive research contributes to the quality of both the process and outcomes of research…’ I think this is a significant statement to underpin the rationale for this study. I would prefer if the researchers elaborate more on this point why this type of research contributes to the quality of research processes and outcomes.

Method

  • Is ethical approval obtained from an ethical board to execute this research? Please add this information to the procedure description.
  • In the first paragraph of the method section the definition of action-oriented research is described. Can the researchers please describe in more detail how the action-oriented research was used?
  • In the section of the ‘study design’ it is explained that data was collected through participant observations. In the second paragraph of this section, it was written that ‘The research duo also wrote reflections…’ I thought these reflections were based on the participant observations, or did I get it wrong here? Did the participant observations resulted in another type of data?
  • The individual reflections, which were written in the diaries, were based on the research questions. What was the starting point for the other types of data (the blogs, vlogs and participant observations)? Was there a framework or something like that?
  • What was the rationale to use both the diaries and the blogs and vlogs? As both types of data reflect on experiences short after they took place.

Results

  • Theme 2, self-awareness and competence-building, seems mainly based on the experiences and reflections of researcher 2. Where there no substantial insights regarding this theme from researcher 1?
  • Theme 3: what is the overarching message of this theme? It seems that other types of communication are needed (is this always the case or was that just because of the researchers’ visual impairment?), but there is also information about making the research results more widely accessible, which I think is about another level of communication
  • Theme 6: line 344: which researcher is meant? And the name of one the researchers is mentioned in line 347.

Discussion/conclusion

  • The results presented in this paper are based on the experiences and reflections of two researchers. Inevitably, future research is recommended. Although the suggestions that are made in this section seem to relate to other (more overarching) themes than this study revealed. Could you add a more specific suggestions about how these results can be underpinned (validated) within future research?
  • In the limitation section the currant way of funding and its shortcomings is mentioned. Is this a limitation of this study? Or is this an advice to rearrange the way of funding based on your results?

Author Response

Dear Reviewer,

Thank you very much for your supportive words and compliments. We found your comments extremely helpful and have revised accordingly. Attached you find a document indicating exactly how we addressed each concern or problem and describing the changes we have made. The changes are marked in red in the paper.

Sincerely,

In name of all authors,

Sofie Sergeant

Reviewer 2 Report

Please refer to attached file.  Comments provided in EASY READ.

Author Response

(The authors gave the same response as above.)

Reviewer 3 Report

The article presents a very valuable research work. It is certainly necessary that this type of work could be published to feed academic debates on inclusive research. It has been a pleasure to review this article.

From my point of view, attention should be paid to the following issues, with the aim of improving the rigorousness of a scholarly and research article such as this one:

  • Although a previous paper describing the training process is referenced in line 72-73, a brief paragraph briefly describing such training should be included in this article.
  • Lines 155-159: Incorporate some elements (examples and excerpts from the minutes/recordings of these discussion processes preferably) that, by way of example, served to enhance the use of these diaries as research strategies (reflection and data collection).
  • In relation to the thematic coding processes (data analysis) the authors should provide evidence (or at least explain in some depth) to what extent the relationships that are established promote that the gaze (p.o.v.) of academic researchers is respectful of the analysis and selection of relevant events/data carried out by researcher 2 (Person with intellectual disability). Lines 190-196.What elements were most difficult in this task, and how do you ensure that the academic's point of view does not silence that of the non-academic? Illustrate with examples from field diaries would be nice.
  • In the Findings section you should highlight, in a different format - italics, quotation marks... - all verbatim quotations and include below, in parentheses, who said them or from which field diary they came from. For example, lines 228-231. This is very important throughout the whole article. Otherwise, it seems that no attention has been paid to the data collection or documentation process. My impression is that the researchers have taken great care of this aspect over the years of work (well done!) and now they cannot so ambiguously pose this data.

Author Response

Dear Reviewer,

Thank you very much for your supportive words and compliments. We found your comments extremely helpful and have revised accordingly. Attached you find a document indicating exactly how we addressed each concern or problem and describing the changes we have made. The changes are marked in red in the paper.

Sincerely,

all authors

Round 2

Reviewer 3 Report

Thank you for your effort. I am willing to see this paper published